# Laser-excited elastic guided waves reveal the complex mechanics of nanoporous silicon

Marc Thelen [1,6], Nicolas Bochud[2,6], Manuel Brinker [1], Claire Prada[3] & Patrick Huber [1,4,5✉]

Nanoporosity in silicon leads to completely new functionalities of this mainstream semiconductor. A difficult to assess mechanics has however significantly limited its application in fields ranging from nanofluidics and biosensorics to drug delivery, energy storage and photonics. Here, we present a study on laser-excited elastic guided waves detected contactless and non-destructively in dry and liquid-infused single-crystalline porous silicon. These experiments reveal that the self-organised formation of 100 billions of parallel nanopores per square centimetre cross section results in a nearly isotropic elasticity perpendicular to the pore axes and an 80% effective stiffness reduction, altogether leading to significant deviations from the cubic anisotropy observed in bulk silicon. Our thorough assessment of the wafer-scale mechanics of nanoporous silicon provides the base for predictive applications in robust on-chip devices and evidences that recent breakthroughs in laser ultrasonics open up entirely new frontiers for in-situ, non-destructive mechanical characterisation of dry and liquid-functionalised porous materials.

[1] Hamburg University of Technology, Institute of Materials and X-Ray Physics, Hamburg, Germany. [2] MSME, CNRS UMR 8208, Univ Paris Est Creteil, Univ Gustave Eiffel, Creteil, France. [3] Institut Langevin, ESPCI Paris, Université Paris Sciences et Lettres, CNRS, Paris, France. [4] Deutsches Elektronen-Synchrotron DESY, Centre for X-Ray and Nano Science CXNS, Hamburg, Germany. [5] Hamburg University, Centre for Hybrid Nanostructures CHyN, Hamburg, Germany. [6]These authors contributed equally: Marc Thelen, Nicolas Bochud. ✉email: patrick.huber@tuhh.de

Exploitation of new "nano"-physical effects of matter confined in nanoporous media, like novel phase behaviour and transport properties, is already beginning to revolutionise technologies ranging from water filtering[1] and drug delivery[2] to energy harvesting, transformation and storage[3–6]. In particular nanoporous silicon (pSi) experiences here an unbroken interest from different fields of research and applications. It provides a monolithic single-crystalline medium with anisotropic pores for the fundamental study of confinement effects on matter[7–13]. Interesting optical, electrical and thermal properties attract the attention of the applied sciences in fields like optoelectronics[14], biosensing[15,16] thermoelectronics[17], mechanical actuation[18,19] and energy storage[3,4]. Applications in microelectromechanical systems (MEMS) are promising due to the great availability and compatibility of pSi: The raw material silicon is one of the most common elements on earth and available in unparalleled qualities. Wafer-scale pSi can be integrated into existing electrical circuits with reasonable expense since bulk silicon is the predominant material for semiconductor devices[16,20]. Furthermore, pSi is bio-compatible and can be functionalised, by changing, extending or enhancing its properties by subsequent treatments, by incorporating additional functional materials[2–4,7,19,21] or direct laser writing in pore space[22].

A scaffold structure of self-organised, tubular pores in pSi can be obtained by etching single-crystalline silicon electrochemically[7,16]. Thereby, the effective stiffness of the material is drastically reduced. Notwithstanding, for many applications the stiffness remains a critical factor, which has scarcely been addressed. The assessment of the mechanical properties of porous systems, like pSi, is challenging with frequently applied, traditional methods owing to the complexity of its crystalline structure (i.e., anisotropy) and geometry (i.e., flat and weak membrane). For example, techniques like indentation, which has typically been used to investigate the elastic properties of pSi under the assumption of mechanical isotropy[7,23], underestimates the complexity of the material by neglecting the influence of the pores' orientation, their geometry and the crystalline matrix[17]. The most commonly applied methods, including conventional ultrasound techniques such as pulse-echo or through-transmission (in the MHz-range), acoustic microscopy (in the GHz-range)[24,25] and laser ultrasonics[26], are in principle capable of a complete characterisation of anisotropic systems. These however require measurements of bulk waves in different crystallographic directions[27], which evidently is problematic for thin porous membranes. Therefore, these methods generally only reported values for the longitudinal bulk wave velocity in the thickness direction[28,29]. Furthermore, most of these methods typically use a coupling medium between the transducers and the sample under investigation. Due to the high capillary pressure in the porous network of pSi, the fluid would be imbibed into the pores and change the pristine mechanical properties. In addition, the contamination of the pores by the fluid limits further investigations of the sample. Yet other techniques, like Brillouin scattering and inelastic neutron scattering, probe the stiffness only on a very local level. For many materials, the penetration depth in Brillouin scattering is very limited[30]. Furthermore, the determination of bulk velocities of highly porous samples is not possible[31]. Inelastic neutron scattering probes samples on the atomic scale, giving valuable insights into the stiffness of the matrix of porous materials but lacking information on the effective elastic properties[17].

Over the past decades, elastic guided waves (EGW) have turned into the focus of the non-destructive testing and structural health monitoring communities. EGWs have first been described by Horace Lamb at the beginning of the nineteenth century[32] and can be observed in plates and shells with a thickness of the order of the bulk wavelength. EGWs offer several advantages to mechanically probe anisotropic and porous samples at the mesostructural scale over traditional methods. Their dispersive and multimodal nature is particularly attractive when it is necessary to measure in-plane elastic properties, as the components of the elastic tensor affect each guided mode differently and with different sensitivities[33]. Waveguide characteristics, such as thickness, stiffness or porosity, can then be deduced from measurements by fitting a waveguide model to the experimental data. This stage, referred to as the resolution of an inverse identification procedure, consists in adjusting the parameters of the model until the modelled and measured guided modes are matched[34].

Some guided modes of higher order exhibit a remarkable behaviour at frequencies where the group velocity vanishes while the phase velocity remains finite. At these zero group velocity (ZGV) frequencies, sharp and local resonance effects are observed. The sensitivity of the ZGV modes has been exploited for precise local plate thickness measurements[35], for absolute and local measurements of the Poisson's ratio of several isotropic media[36,37], or to highlight the anisotropic constitutive behaviour of bulk silicon[38]. Recent studies also reported on the mechanical characterisation of complex nanoscale structures such as low-density nanoporous gold foams[39] or nanoscale bilayers consisting of a silicon-nitride plate coated with a titanium film[40].

In this work, we propose a thorough assessment of the effective mechanical behaviour of pSi by exploiting the characteristics of EGWs and ZGV modes combined with waveguide modelling and inverse identification. The waveguide characteristics are measured by a laser ultrasonics (LUS) technique: waves are excited by a laser source through thermoelastic conversion and surface displacement is detected by laser interferometry (see Fig. 1). Besides the anisotropic stiffness tensor evaluation, the influence of the pore conicity on the effective mechanical behaviour will be examined. Cone-shaped pores and their significant influence on key properties like optical response and capillary pressure have already been evidenced for pSi[41–44]. A model has recently been developed to describe the pore depth-dependency in capillary rise experiments[45]. In order to outline the impact of the pore network, pSi is first compared to bulk silicon. Measurements and modelling results are then discussed for dry and liquid-infused pSi, highlighting functionalisation induced changes in the mechanical behaviour and emphasising the feasibility and accuracy of the proposed approach for characterising nanoporous systems.

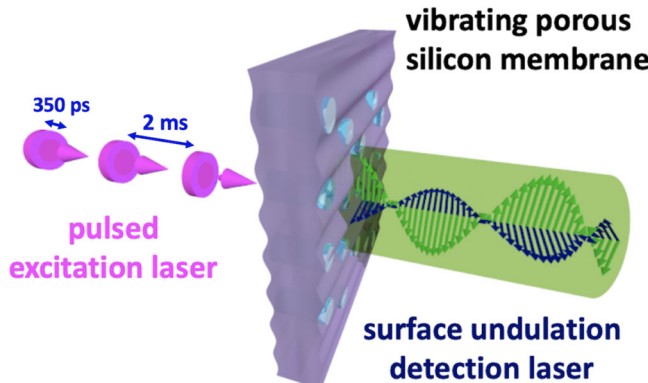

**Fig. 1 Laser ultrasonics experiment.** Laser-induced thermoelastic excitation of elastic guided waves in a nanoporous silicon membrane along with laser-interferometrical detection of the resulting undulations dynamics at the membrane surface. Shown is a symmetric elastic guided wave.

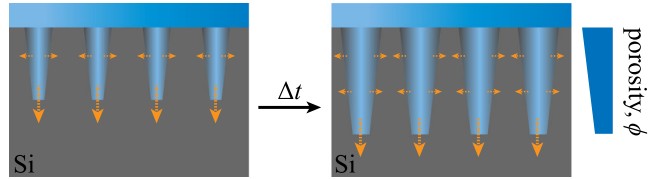

**Fig. 2 Pore widening in nanoporous silicon.** Isotropic pore widening during electrochemically etching silicon, which causes the depth-dependent porosity variation of nanoporous silicon.

## Results

**Investigated samples.** A free-standing 105-μm-thick pSi membrane was synthesised by electrochemical etching. In general, a porosity of around 50% and a mean pore radius of $37 \pm 2$ Å, determined by the nitrogen sorption isotherm method, are expected for samples synthesised with the process parameters described in "Methods". In this study, a porosity of $\phi = 55 \pm 1\%$ for the synthesised sample was estimated by weighting and volume measurement. The slight difference between the two techniques is explained by small deviations in the synthesis.

Moreover, it is hypothesised that during the synthesis of pSi the pore walls exposed to hydrofluoric acid (HF) were isotropically etched[41], although quantum confinement in the pore walls is likely to inhibit the attack of HF[46]. The upper part of the membrane was exposed longer to the acid and thus cone-shaped pores developed (Fig. 2). This assumption is supported experimentally, as samples with increasing thickness typically highlight a broader mean pore distribution due to longer exposure. For instance, a mean pore radius of $61 \pm 6$ Å was determined for a $277 \pm 6$-μm thick sample.

The membrane was first investigated empty and subsequently filled with liquid. The purpose of the liquid was to vary the filling fraction of the pores. Changes in the wave's velocities can be traced back to the variation in filling fractions, as they are inversely proportional to the square-root of the effective density. As there was no controlled environment while measuring the sample, a liquid with a low vapour pressure was required to minimise evaporation. Squalane ($C_{30}H_{62}$) with a density of $\rho_L = 0.807$ g cm$^{-3}$ was chosen as a liquid, since it has a low vapour pressure of $1.5 \pm 0.5$ μPa[47]. The membrane was filled with the liquid by imbibition. Squalane was simply dripped onto the surface. Extant liquid was removed from the surface. The high capillary pressure led to a high filling fraction of around ~80%, which was determined by gravimetric measurement.

A $p^+$ doped bulk silicon wafer with a thickness of around $397 \pm 3$ μm was investigated for comparison. Both samples were coated with a 20 nm thin layer of silver to enhance the interaction with the laser-based measurement setup.

The following sections present a detailed analysis of the measurements and the effective mechanical behaviour of pSi. A short recap of the main characteristics of EGWs and the LUS setup is provided in "Methods".

### Comparison of EGWs in bulk and porous silicon. 

Figure 3 shows the obtained LUS measurements for bulk silicon (upper panels, a–c) and dry pSi (lower panels, d–f). Spatio-temporal broadband signals are depicted in Fig. 3a, d. For both samples, waveforms were acquired along the [110] direction during $t = 6$ μs and averaged 1000 times to improve the peak-to-noise ratio. The laser excitation takes place at $x_1 = 0$ mm (see "Methods"). There the dominant feature is a low-frequency oscillation around 0.5 MHz, with a time-dependent broadening characteristic of EGWs dispersion. Before the low-frequency oscillation wavefronts of higher frequencies can be spotted in the submicroseconds

range. As can be observed, the slope of these wavefronts are much steeper for pSi than for bulk silicon, thus indicating lower wave velocities for the porous sample.

A more advanced insight is given by the dispersion curves obtained by 2D Fourier transform of these spatio-temporal signals[48] and presented in the normalised wavenumber-frequency plane ($kd$–$fd$) in Fig. 3b, e. Prior to Fourier transformation the signals have been filtered by applying apodization (Hanning) windows in both dimensions (time and distance) to avoid secondary lobes[49]. As expected, the reduced cut-off frequency-thickness products for pSi confirm that the longitudinal ($V_L$) and shear ($V_T$) bulk wave velocities in the [001]-plate thickness direction are lower compared to those of bulk silicon, which is attributed to a concurrent reduction of both the stiffness coefficients in that direction and the mass density.

A closer look on the two dispersion spectra reveals further differences (highlighted by dashed ellipses). The zero-order symmetric ($S_0$) and antisymmetric ($A_0$) modes for bulk silicon cross each other and converge to the Rayleigh wave velocity at $kd \approx \pm 7$. In contrast for the pSi sample, the two lowest order modes repulse each other at $kd \approx \pm 8$. Such repulsion can be observed for higher-order modes too. For instance, $S_1$ and $A_1$ modes cross each other near the first ZGV resonance for the bulk silicon but the corresponding modes repulse each other for pSi. Note that similar repulsion effects already have been observed for other asymmetric structures, such as bilayers[50,51] or functionally graded waveguides[52]. Strictly speaking, the labelling of the modes introduced in "Methods" for bulk silicon is no longer valid for pSi. Therefore, these modes are now considered to be quasi-symmetric ($qS$) or quasi-antisymmetric ($qA$).

Overall, this behaviour is traced back to a break in symmetry in the mid-plane of the membrane, caused by the shape of the directional pores. Owing to this symmetry reduction the two lowest order modes now diverge, each one with a different Rayleigh wave velocity associated with a propagation path on a different surface. Concretely, the $qS_0$ mode converges toward the faster Rayleigh wave on the dense surface and the $qA_0$ mode toward the slowest Rayleigh wave on the light surface. In Fig. 3e, only the $qA_0$ mode is observed because our measurements were performed on the sample face with larger pore openings. This effect is discussed further in the Supplementary Fig. 1.

The frequency spectra in Fig. 3c, f were obtained by temporal Fourier transform of the signal measured with the laser source and probe on epicentre. They are composed of two parts: a spread spectrum in the low-frequency range, corresponding to the flexural mode (i.e., $A_0$ or $qA_0$), and several sharp peaks corresponding to ZGV resonances (dashed red lines). As can be observed, the three resonances measured in bulk silicon display a splitting into two peaks (inserts of Fig. 3b, c), which is a signature of the sample's anisotropy[38]. The excitation was indeed performed with a point source, thus triggering modes in all propagation directions. For instance, the second ZGV resonance frequency ranges from 7.94 MHz.mm for the [110] propagation direction (denoted by red dots) to 7.88 MHz.mm for the [100] propagation direction (denoted by black dots). So, after enough time for the propagating modes to escape from the measured area, at each frequency, this sharp ZGV resonance dominates the signal and corresponds to a plane stationary wave (see Supplementary Fig. 2). This leads to an additional pseudo cut-off frequency. In contrast to the bulk silicon sample, for pSi, a splitting of the ZGV resonances cannot be clearly resolved (see the enlargement of Fig. 3f). This observation suggests that, although some degree of anisotropy may remain in the $x_1x_2$-plane of the plate, for the considered porosity the presence of the pores prevails over the original crystalline nature of bulk silicon (i.e., cubic anisotropy)[19,53].

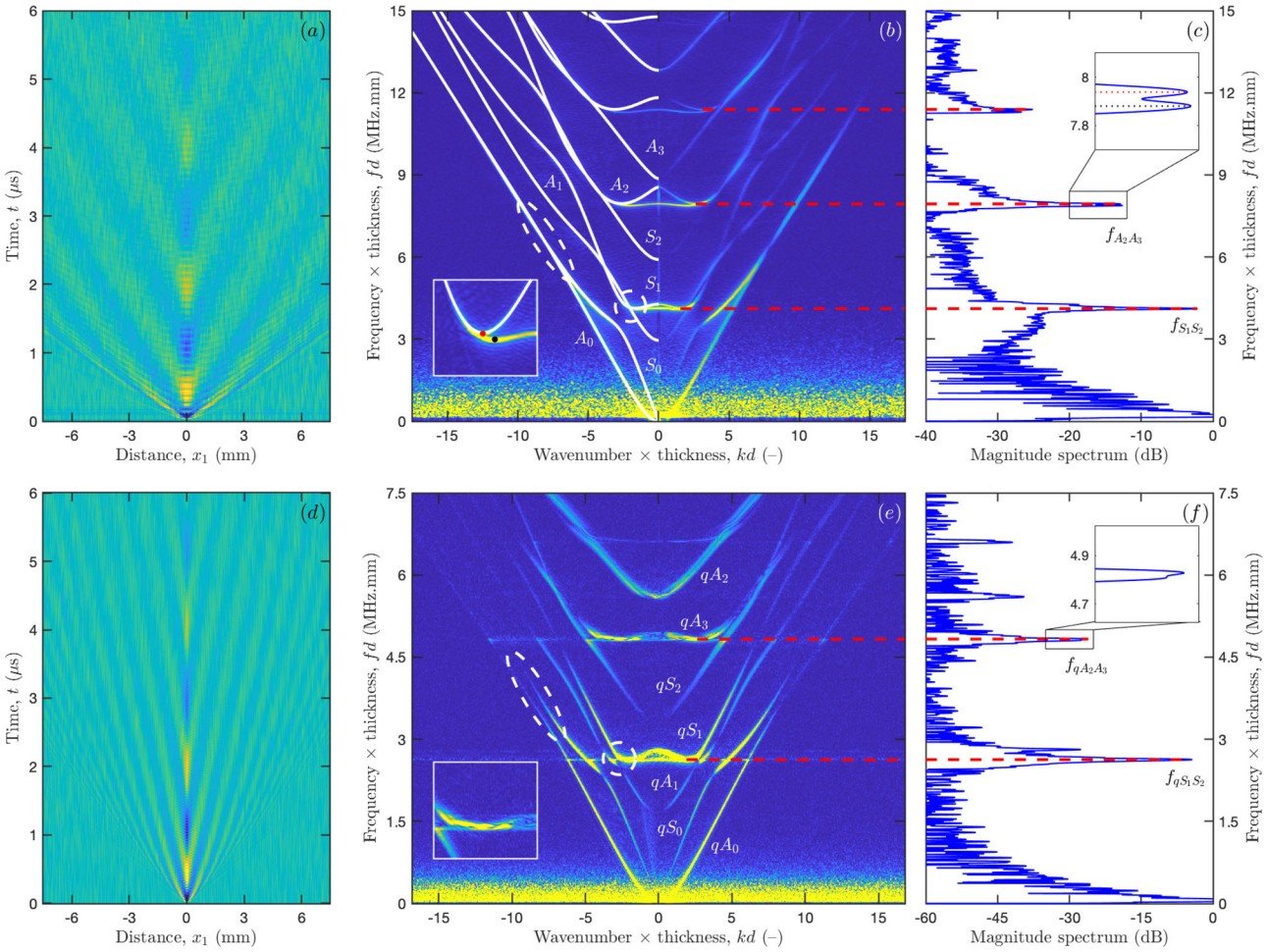

**Fig. 3 Laser ultrasonics measurements for bulk silicon (upper panels) and nanoporous silicon (lower panels). a–d** Spatio-temporal broadband displacements measured on the samples with the laser ultrasonics setup; **b–e** Dispersion curves in the normalised wavenumber-frequency plane, obtained by 2D Fourier transform of the spatio-temporal signals (for bulk silicon, theoretical guided modes are displayed in white continuous lines for comparison); and (**c–f**) zero group velocity (ZGV) resonances (the insert depicts the presence of two ZGV resonances for bulk silicon, where the red one is the actual resonance belonging to [110]-axis and the black one reveals the concurrent excitation of the resonance belonging to [100]-axis due to in-plane anisotropy).

For bulk silicon, the measured guided modes are in excellent agreement with the guided modes predicted by the theory (continuous white lines in Fig. 3b, see "Methods"). To retrieve quantitative information from the measured guided modes in pSi and account for the aforementioned features (e.g., modes repulsion, reduced properties, nature of liquid into the pores), a custom model-based approach is applied in next section.

**Assessing the effective mechanical behaviour of porous silicon.** In what follows, we briefly introduce the waveguide model that is used to capture the effective mechanical behaviour of pSi (dry and liquid-infused), as well as the objective function and model parameters sweeping routine involved in the inverse identification procedure used to match measured and predicted guided modes.

**Forward modelling of guided modes in pSi.** pSi was modelled as a 2D transverse isotropic homogeneous free plate waveguide with linearly varying stiffness-to-weight ratios through the thickness. Several observations guided the choice of this model. First, in the frequency bandwidth of interest (several tens of MHz), the wavelengths of EGWs, ranging from around 100 μm to 10 mm, are much larger than the typical size of the heterogeneities (nanometric pores). pSi can thus be considered as a homogeneous propagation medium. Second, based on experimental observations

(Fig. 3e–f, we hypothesise that the porous network prevails over the crystalline nature of the bulk silicon membrane, so that the mechanical behaviour of pSi can be approximated by a transversely isotropic waveguide model ($x_1 x_2$ being the isotropy plane). Consequently, its effective mechanical behaviour can be described by means of four independent stiffness coefficients in 2D, i.e., $c_{11}$, $c_{33}$, $c_{13}$, and $c_{55}$[34]. Third, the depth-dependent porosity $\phi(x_3)$ (recall Fig. 2) turns into continuously varying stiffness coefficients $c_{ij}(x_3)$ and density $\rho(x_3)$ across the thickness. Since the competing impact of stiffness versus density cannot be easily disentangled, we introduce a function $\alpha(x_3)$, which is driven by a factor $\beta$ that accounts for the cone-shaped pores:

$$\alpha_{ij}(x_3) = \frac{c_{ij}(x_3)}{\rho(x_3)} = 2\alpha_{ij}^0 \left( (1 - 2\beta) \frac{x_3}{d} + \beta \right), \quad (1)$$

where $\alpha(x_3)$ represents a linearly varying anisotropic stiffness-to-weight tensor across the thickness, with $\alpha^0$ being the stiffness-to-weight tensor in the mid-plane of the membrane, i.e., at the position $x_3 = d/2$. It should be noted that $\alpha_{ii}(x_3)$ corresponds to the square of the bulk wave velocities, meaning that, overall, Eq. (1) accounts for continuous variations of bulk wave velocities through the membrane's thickness. Note also that setting $\beta = 0.5$ leads to the trivial case of cylindrical pores, and therefore to constant bulk wave velocities across the thickness. The density in

**Table 1 Bounds of the model parameters Θ and their respective discretisation used to perform the inverse identification.**

| pSi | | Stiffness-to-weight ratios | | | | Conicity | Density |
|---|---|---|---|---|---|---|---|
| | | $\alpha_{11}^{0}$ | $\alpha_{33}^{0}$ | $\alpha_{13}^{0}$ | $\alpha_{55}^{0}$ | $\beta$ | $\rho_{\mathrm{P}}$ |
| | | Stiffness-to-weight ratios (mm² μs⁻²) | | | | – | g cm⁻³ |
| Dry | Range | 19–23 | 29.3–32.5 | 6.6–9.6 | 10.2–11.4 | 0.5–0.65 | 0 |
| | Step | 1 | 0.8 | 0.5 | 0.3 | 0.025 | – |
| Liquid-infused | Range | $\dfrac{\hat{c}_{11}(d/2)}{\rho(d/2)}$ | $\dfrac{\hat{c}_{33}(d/2)}{\rho(d/2)}$ | $\dfrac{\hat{c}_{13}(d/2)}{\rho(d/2)}$ | $\dfrac{\hat{c}_{55}(d/2)}{\rho(d/2)}$ | 0.6 | 0.6–1 |
| | Step | – | – | – | – | | 0.025 |

the mid-plane of the membrane can be defined as

$$\rho(d/2) = \rho_{\mathrm{m}}\big(1 - \phi(d/2)\big) + \rho_{\mathrm{p}}\phi(d/2), \qquad (2)$$

where $\rho_{\mathrm{m}}$ and $\rho_{\mathrm{p}}$ are the density of the membrane's matrix and of the material within the pores (dry or liquid-infused pSi), respectively. In this way, the stiffness coefficients in the mid-plane can easily be derived as $c_{ij}(d/2) = \rho(d/2)\alpha_{ij}^{0}$.

The dispersion characteristics of pSi were investigated using an approach based on Bloch–Floquet analysis[54]. The numerical model was defined in terms of a unit cell consisting of a rectangular domain, whose side length and thickness amount to $a = 10\,\mu m$ and $d = 105\,\mu m$, respectively. The constitutive material properties and the stiffness-to-weight tensor of the unit cell were set according to the values reported in Table 1 and Eq. (1), respectively. Periodic Bloch–Floquet conditions were imposed as displacement conditions on the lateral boundaries of the unit cell, whereas the remaining boundaries were considered as traction-free. Finally, a wavenumber vector **k**, ranging from 0 to 160 mm⁻¹ (thus satisfying the regime associated with the irreducible Brillouin zone), was imposed and the corresponding frequencies **f** were retrieved by solving the corresponding eigenvalue problem. This model has been implemented using the commercial software COMSOL Multiphysics®.

To serve as an example, Fig. 4 depicts typical guided modes for dry pSi (i.e., $\rho_{\mathrm{p}} = 0\,g\,cm^{-3}$) modelled using factors accounting for the pores' conicity equal to $\beta = 0.575$ (continuous black lines) and $\beta = 0.5$ (dashed red lines). As can be observed, unlike for bulk silicon (see Fig. 7 in "Methods") and pSi with cylindrical pores ($\beta = 0.5$), guided modes of different symmetry cannot cross each other for pSi with cone-shaped pores ($\beta = 0.575$), thus consistently capturing the experimental evidences observed in Fig. 3e. This is explained by the linearly varying stiffness-to-weight tensor $\alpha(x3)$ across the thickness, which breaks the plate symmetry. In particular, the lowest order modes ($qA_0$) et ($qS_0$) do not converge to the same slope for large wavenumbers $k$, thus resulting in Rayleigh waves with different velocities on each membrane surface at high frequencies.

**Inverse identification of the model parameters**. In this section, we propose an inverse identification procedure to infer the model parameters that yield the best agreement between the measured guided modes and the guided modes derived from the parametric forward modelling of pSi. Such multiparametric model-based approach requires the definition of an objective function and its optimisation based upon an automatic search routine.

The objective function $F(\boldsymbol{\theta})$ is defined as the mean value of the element-wise product between experimental and modelled dispersion maps,

$$F(\boldsymbol{\theta}) = \frac{1}{PQ}\sum_{p=1}^{P}\sum_{q=1}^{Q} D_{pq}M_{pq}(\boldsymbol{\theta}) \qquad (3)$$

where $\boldsymbol{\theta}$ is a vector that contains the model parameters. The

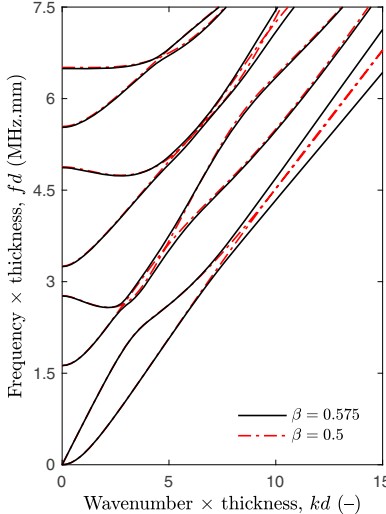

**Fig. 4 Modelling of elastic guided modes.** Dispersion curves in the normalised ($kd$, $fd$)-plane calculated using Bloch–Floquet analysis for a dry nanoporous silicon membrane of thickness $d$: cone-shaped pores ($\beta = 0.575$) vs cylindrical pores ($\beta = 0.5$).

experimental dispersion map $D$ is obtained by applying a threshold to the measured dispersion curves (e.g., Fig. 3e), in order to force the noisy background to be zero (so that the resulting matrix is sparse). The modelled dispersion map $M(\boldsymbol{\theta})$ is obtained by converting the modelled guided modes into a binary image that has the same dimensions $P \times Q$ than $D$.

Formally, the optimal model parameters $\hat{\boldsymbol{\theta}}$ result from the maximisation of the objective function $F(\boldsymbol{\theta})$ as,

$$\theta = \underset{\boldsymbol{\theta} \in \Theta}{\arg\max}\, \boldsymbol{F}(\boldsymbol{\theta}) \qquad (4)$$

where $\Theta$ denotes the bounds of the model parameters $\boldsymbol{\theta}$. The objective function $F(\boldsymbol{\theta})$ was calculated for different model parameters $\boldsymbol{\theta}$ along a multidimensional grid in steps[55], according to the values reported in Table 1.

The porosity of pSi in the mid-plane of the membrane and the density of its matrix phase were assumed to be known and set to $\phi(d/2) = 55\%$ and $\rho_{\mathrm{m}} = 2.329\,g\,cm^{-3}$, respectively. For dry pSi, the bounds of the model parameters $\Theta$ were roughly estimated by considering the two first ZGV resonance frequencies from Fig. 3f, i.e., $f_{qS_1S_2}d = 2.628\,MHz.mm$ and $f_{qA_2A_3}d = 4.827\,MHz.mm$ (see "Methods"). Following ref. [36], the ratio $f_{qA_2A_3}/f_{qS_1S_2}$, respectively, yields longitudinal and shear bulk wave velocities of $V_{\mathrm{L}}^{\perp} = 5.54$ mm μs⁻¹ and $V_{\mathrm{T}} = 3.26$ mm μs⁻¹ in the plate thickness direction. In addition, a rough estimates of the longitudinal bulk wave velocity in the plane of the plate $V_{\mathrm{L}}^{\parallel} = 4.55$ mm μs⁻¹ was derived from the phase velocity of the nondispersive part of the ($qS_0$) mode (Fig. 3e),

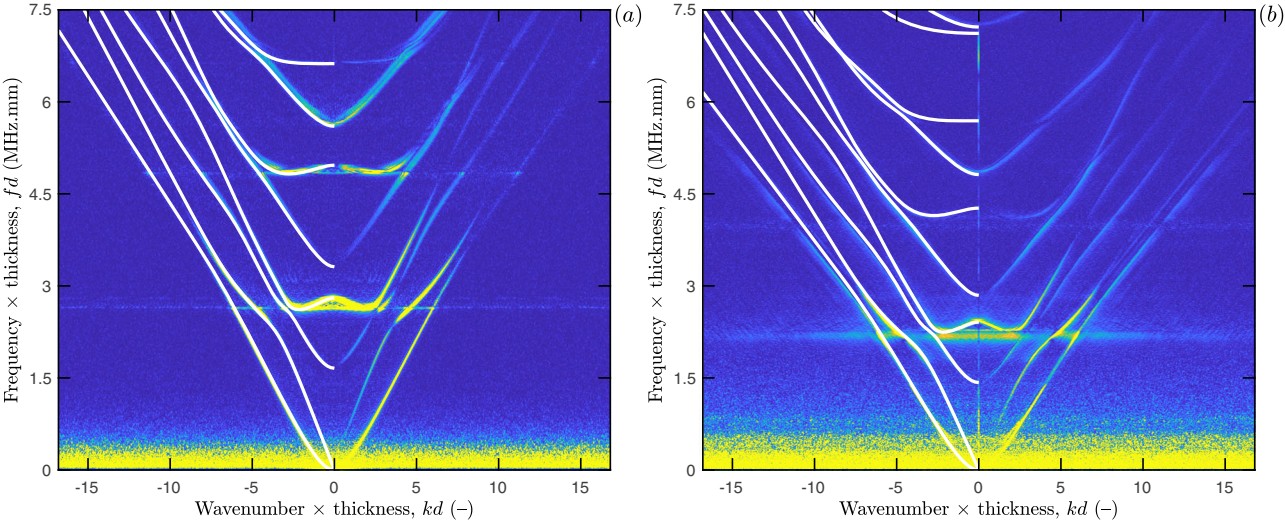

**Fig. 5 Inverse problem solutions.** Optimal matching between the measured and modelled guided modes for (**a**) dry and (**b**) liquid-infused nanoporous silicon.

which can be referred to as the plate wave speed[56,57],

$$V_P = 2V_T \sqrt{1 - \left(\frac{V_L^{\parallel}}{V_T}\right)^2}. \qquad (5)$$

These three bulk wave velocities in turn lead to stiffness-to-weight ratios of $\alpha_{11}^0 = 20.7$ mm$^2$ μs$^{-2}$, $\alpha_{33}^0 = 30.7$ mm$^2$ μs$^{-2}$, and $\alpha_{55}^0 = 10.6$ mm$^2$ μs$^{-2}$ in the mid-plane of the membrane. The ranges reported in Table 1 were thus selected to be around ±5% of these approximated values. The range for the out-of-diagonal ratio $\alpha_{13}^0$ was empirically set equal to 6.6–9.6 mm$^2$ μs$^{-2}$. The range for the factor $\beta$, which accounts for the cone-shaped pores, was selected to encompass values for a strong modes repulsion and the baseline corresponding to cylindrical pores (recall Fig. 4). The grid steps were defined based on a trade-off between identification error and computation time. The multiparametric inverse approach was first solved for dry pSi. Assuming that the liquid mostly impact the effective density, the resulting optimal model parameters $\hat{\boldsymbol{\theta}}$ were then used to match liquid-infused pSi, simply by adjusting the density $\rho_p$ of the medium within the pores in Eq. (2), whose range was chosen to encompass values around ±25% the reference filling fraction for squalane.

The results obtained for the initially empty and subsequently filled pSi sample are displayed in Fig. 5. For the dry sample, the optimal matching between the experimental data and the guided modes predicted by the proposed modelling approach shows an excellent agreement. The optimal model parameters $\hat{\boldsymbol{\theta}}$ estimated using the inverse identification procedure were as follows: $\hat{\alpha}_{11}^0 = 21$ mm$^2$ μs$^{-2}$, $\hat{\alpha}_{33}^0 = 31.7$ mm$^2$ μs$^{-2}$, $\hat{\alpha}_{13}^0 = 7.6$ mm$^2$ μs$^{-2}$, $\hat{\alpha}_{55}^0 = 11.1$ mm$^2$ μs$^{-2}$, and $\hat{\beta} = 0.6$, which in turn lead to the following stiffness coefficients $\hat{c}_{11}(d/2) = 22$ GPa, $\hat{c}_{33}(d/2) = 33.2$ GPa, $\hat{c}_{13}(d/2) = 8$ GPa, and $\hat{c}_{55}(d/2) = 11.6$ GPa in the mid-plane of the membrane. These values were then used to optimise the liquid-infused sample model, leading to an optimal fluid density within the pores equal to $\hat{\rho}_p = 0.675$ g cm$^{-3}$. As can be observed, in this case the matching between the measured and modelled guided modes is again remarkable up to a wavenumber-thickness product $kd \approx 7$ but only moderate for larger wavenumbers, i.e., smaller wavelengths. This point will be further addressed in the discussion below. Moreover, the inferred value for the optimal fluid density $\hat{\rho}_p$ corresponds to a filling fraction of 84%, which is in good agreement with the reference gravimetric measurement of ~80%. The resulting variations of bulk

wave velocities through the membrane's thickness for both the dry and liquid-infused sample are reported in Supplementary Fig. 3.

## Discussion
This study investigated the ability of EGWs to characterise the effective mechanical behaviour of pSi using a non-contact and non-destructive LUS setup. The main findings from this study are as follows: (1) our technique allows measuring multiple guided modes and sharp ZGV resonances in thin nanoporous membranes and it is sensitive to the presence of fluid inside the pores; (2) our measurements evidenced that the synthesis of pSi results in an asymmetric membrane, whose impact manifests itself as a repulsion of the guided modes. Moreover, this repulsion effect could be adequately modelled by considering a linearly varying stiffness-to-weight tensor across the membrane's thickness, which is traced back to the cone-shaped geometry of the pores; (3) the porous network was assumed to prevail over the crystalline nature of the bulk silicon membrane, and in such a case using a transversely isotropic model was proven to represent a reasonable approximation to identify waveguide properties at the mesostructural scale; and (4) the resulting effective stiffness coefficients $\hat{c}_{ij}(d/2)$ in the mid-plane of the membrane are much lower than those of bulk silicon (reduction of ~80%).

Furthermore, one can also build upon the former results to extrapolate the stiffness coefficients of the matrix phase $c_{ij}^m$ of pSi using asymptotic homogenisation[58]. In this modelling approach initially devoted to cortical bone[59], pSi is represented as a two-phase composite material made of a homogeneous matrix with cubic anisotropy pervaded by periodically distributed cylindrical pores. This structure leads to transversely isotropic elasticity at the mesoscale with the following stiffness coefficients of the silicon matrix: $c_{11}^m = 80$ GPa, $c_{12}^m = 30$ GPa, and $c_{44}^m = 40$ GPa. It is worth pointing out that these values are ~50% lower than those typically reported for bulk silicon. This reduction can be traced back to inactive silicon walls as discussed in ref. 19. In that study, the authors modelled the pSi network based on transmission electron micrographs, therefore taking its irregular microstructure into account, and thus providing a more accurate description of the sensitive influence of the silicon scaffold structure on the mechanics compared to earlier work based on an idealised regular honeycomb structure[18,60]. A bulk-like elasticity in the silicon walls is also supported by recent inelastic neutron scattering experiments on mesoporous silicon[61], which probe

locally, on the single-pore-wall scale the mechanics and are thus not affected by mesoscale defects in the silicon scaffold structure. These measurements indicate that the phonon group velocities in nanoporous silicon are not modified by nanostructuring down to sub-10 nanometre length scales and no evidence can be found for phonon-softening in topologically complex mesoporous silicon putting it in contrast to silicon nanotubes and nanoribbons.

A direct quantitative comparison with earlier reported bulk wave velocities is difficult, as the synthesis of pSi may differ from one study to another. Indeed, apart from the porosity amount, doping level or applied current intensity and duration, most studies conducted in the ultrasound field considered a pSi layer etched on a bulk silicon substrate and used coupling medium that is likely to be imbibed into the pores, thus potentially resulting in different pore characteristics (e.g., diameter, morphology, fluid content). For instance, using an ultrasound through-transmission setup for measuring a water immersed pSi membrane with a porosity of 50%, Bustillo et al.[29] reported a longitudinal bulk wave velocity of $5.16 \, \text{mm} \, \mu\text{s}^{-1}$ in the sample's thickness direction, which is in moderate agreement with our values for dry (i.e., $5.63 \, \text{mm} \, \mu\text{s}^{-1}$) and squalane-filled (i.e., $4.84 \, \text{mm} \, \mu\text{s}^{-1}$) pSi. Interestingly, Aliev et al.[28] reported an empirical law for the porosity dependent longitudinal bulk wave velocity in the sample's thickness direction, which yields a value of $5.3 \, \text{mm} \, \mu\text{s}^{-1}$ for a porosity of 55%. Their pSi sample was a heavily doped $p^{++}$ membrane and they indicated that the obtained velocities were significantly higher than those previously observed for $p^+$-doped samples[24]. Some authors also extrapolated values for the shear bulk wave velocity, Rayleigh wave velocity or stiffness coefficients[25], but these results have a limited meaning because they all were derived under the assumption of mechanical isotropy.

In this regard, our EGWs-based approach outperforms such traditional methods, as it allows for the concurrent assessment of two longitudinal and one shear bulk wave velocities from a single measurement sequence, as well as a structural parameter related to the shape of the pores. As a further advantage, this non-contact technique opens promising perspectives for the monitoring of the filling process of pSi. Although the overall agreement between measured and modelled guided modes was only moderate for the liquid-infused sample (Fig. 5b), this was excellent up to wave-numbers $k$ encompassing the ZGV resonances. Since these ZGV modes are spatially localised, they may thus allow for spatial mapping of the sample properties. This mismatch, however, is not too surprising since we considered here a simple effective medium model driven by the liquid density only, and thus neglected the (minor) impact of the liquid on the effective stiffness. Other factors explaining this limitation may concern solid-fluid interaction (poro-elasticity) in the light of Biot theory[62,63], in particular liquid rearrangements (viscous flow) owing to unrelaxed pressure conditions or the presence of microcracks[64].

As a limitation, since our measurements are sensitive to wave velocities, the influence of the effective stiffness cannot be decoupled from that of the effective mass density. Therefore, the impact of the cone-shaped pores was accounted for by considering a continuously varying stiffness-to-weight tensor across the thickness, where all components were assumed to vary following the same linear profile. Furthermore, it should be noted that the LUS measurements were performed along the [110]-direction only, thus limiting the investigation of the in-plane anisotropy of the pSi sample. As such, the assumption of a waveguide model with a transversely isotropic mechanical behaviour still demands direct proof, but the obtained inverse results a posteriori justified that this approximation is accurate enough to recover waveguide properties along that direction. Future studies are warranted to address the question of whether or not the in-

plane anisotropy vanishes for pSi. For instance, measuring the dispersion curves along different directions in-plane[33] or tracking the ZGV modes as a function of the line source orientation[38] would allow elucidating whether a certain degree of in-plane anisotropy persists and how much.

Other next steps will be to test our approach on new samples with different porosity amounts and doping levels, as well as different liquids imbibed into the pores, in order to derive porosity- and functionalisation-dependent bulk wave velocity laws. The repulsion phenomenon of the two Rayleigh waves will be further investigated to get a deeper insight in the pores morphology on the upper and lower surfaces of the sample. In addition to the current functionally graded model, a multilayer model could be developed to obtain closed form equations, thereby improving the speed and accuracy of the inverse procedure[33].

In conclusion, in this study we investigated the effective mechanical behaviour of a thin pSi membrane using a non-contact technique based on the propagation of EGWs, in which the waveguide characteristics were measured by a LUS technique. In a first part, we thoroughly discussed the main differences between the measured guided modes in bulk silicon and dry pSi. The main outcomes were that our EGWs and ZGV measurements proved to be very sensitive to the pore conicity that developed during the synthesis process of pSi and that the anisotropy induced by these directional pores prevails over that of the original crystalline nature of bulk silicon. In a second part, we introduced a waveguide model to retrieve quantitative information from the measurements in terms of effective anisotropic stiffness and a surrogate factor related to the shape of the pores. The accuracy of the model, in terms of goodness-of-fit between measured and modelled guided modes, was shown to be excellent to satisfying for dry and liquid-infused pSi, respectively.

This study provided important insights on the mechanics and pore structure of pSi as a very versatile applicable porous medium not only for the fundamental sciences to study confinement effects in matter[6,65,66], but also with regard to applications of self-organised porosity in a mainstream semiconductor with a huge range of technological applications[7,16]. Our study also opens promising perspectives for the in-situ monitoring of the filling process of pSi in a controlled environment with applications in the field of the exploration of the fundamental properties of matter, in particular the peculiar elasticity of liquids in nanopores[67–69] and the interaction of their nanocapillarity with solid elasticity, i.e., elastocapillarity[70–72]. Also the exploration of the peculiar mechanics and plasticity of silicon at the micro- and nanoscale depending on the synthesis method and influence of surfaces[73,74] could be a rewarding undertaking with the technique presented here.

We envision that this contactless technique allows also the versatile combination with other methods, e.g., a simultaneous structural characterisation with x-ray or neutron scattering, while processing under complex sample environments, for examples under vacuum, controlled atmospheres, tempering, or external fields (electrical, magnetical). It will also be of particular value to characterise non-destructively the effective mechanics of other nanoporous materials presently revolutionising materials science. Thus, it will help for a rational design of 3D mechanical robust nanostructured materials, a particular challenge for embedding functional nanocomposites in macroscale devices[75].

## Methods

**Synthesis of porous silicon**. The starting material for the fabrication of the pSi membrane was a $p^+$ doped (001) silicon wafer with a thickness of $525 \pm 25 \, \mu\text{m}$. The resistance of the wafer was in the range of 0.01–0.02 Ωcm. The etchant was a volumetric 4:6 mixture of hydrofluoric acid (HF; 48%, *Merck Emsure*) and ethanol

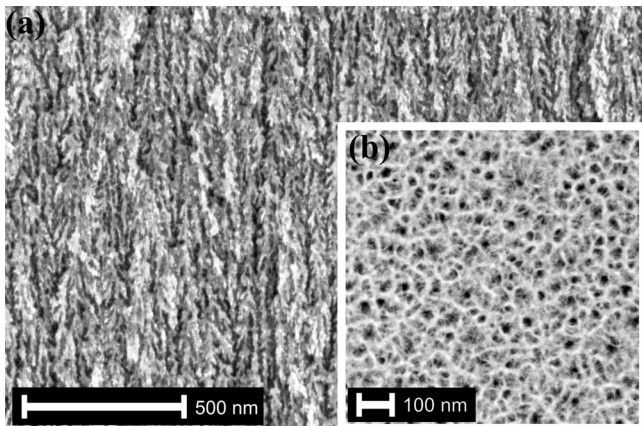

**Fig. 6 Scanning electron microscope images of nanoporous silicon. a** Side view: Cut through the membrane perpendicular to the surface. Dendritically grown pores parallel to [001] branching in small sub-pores with dead ends. **b** Top view: cut through the membrane parallel to the surface. Randomly distributed pores in (001).

(absolute, *Merck Emsure*). A current density of 12.5 mA cm$^{-2}$ was applied between the sample and a platinum counter electrode. After 75 min, a pSi layer with a pore depth of around $d = 105 \pm 3$ μm was obtained. In order to achieve a free-standing membrane, the porous layer was detached by electropolishing. Therefore, a current of 2 A was applied for 30 s. Without any additional pretreatment, this process results in a porous network, with pores randomly distributed in-plane (001) and aligned parallel to [001] (see Fig. 6). The pores have been branched by dendritic growth (see Fig. 6a). After the synthesis procedure, the sample was rinsed with deionised water and dried in the air for one day before the measurements took place.

**Elastic guided waves and resonances**. In this section, we briefly recall the main characteristics of EGWs, together with the different resonances that can occur in anisotropic plates.

The propagation of EGWs is typically represented by a set of dispersion curves in the wavenumber-frequency ($k$–$f$) plane. Guided modes propagating in homogeneous plates are generally categorised into $S$- and $A$-modes, oscillating either symmetrically ($S$) or antisymmetrically ($A$) to the mid-plane of the plate. Modes of the same symmetry cannot cross each other, while a symmetric mode can cross an antisymmetric one. For anisotropic materials, these modes depend on the propagation direction. For instance, Fig. 7 depicts the dispersion curves of the lower order symmetric and antisymmetric modes calculated for a (001)-cut silicon plate using partial wave theory[76]. A silicon crystal of cubic symmetry is characterised by three independent stiffness coefficients in Voigt notation, i.e., $c_{11}$ = 165.6 GPa, $c_{12}$ = 63.9 GPa, $c_{44}$ = 79.5 GPa, and the mass density $\rho$ = 2.329 g cm$^{-3}$ [33]. Dispersion curves were determined for propagation along the principal symmetry axes: <100> (azimuth angle $\gamma = 0°$, in grey) and <110> ($\gamma = 45°$, in black). For non-principal symmetry axes, the dispersion curves of a given mode lie within the bundle delimited by these curves. Note that it is convenient to use variables normalised to the plate thickness $d$, such as the frequency-thickness and wavenumber-thickness products[38].

The lowest order symmetric ($S_0$) and antisymmetric ($A_0$) modes take their origin at zero frequency and converge to a surface wave, known as the Rayleigh wave, when the shear wavelength is small compared to the plate thickness $d$. Higher-order modes originate at a cut-off frequency (i.e., $f_c = f(k=0)$) associated with either the longitudinal ($V_L$) or shear ($V_T$) bulk wave in the [001]-plate thickness direction. For these modes, the group velocity $V_g = d\omega/dk$ vanishes at $k = 0$, giving rise to a thickness resonance at the cut-off frequency (blue dots in Fig. 7). Such thickness resonances can be written according to their symmetry kinds and the parity of their order as

$$\text{Symmetric modes} \begin{cases} S_{2n}: & f_c d = n V_L \\ S_{2m+1}: & f_c d = \frac{2m+1}{2} V_L \end{cases}$$

$$\text{Antisymmetric modes} \begin{cases} A_{2n}: & f_c d = n V_L \\ A_{2m+1}: & f_c d = \frac{2m+1}{2} V_T \end{cases}, \quad (6)$$

where $n \geq 1$ and $m \geq 0$, and $V_L = \sqrt{c_{11}/\rho}$ and $V_T = \sqrt{c_{44}/\rho}$. The subscript associated with each resonance ($S$ or $A$) corresponds to the number of nodes of the mechanical displacement in the thickness of the plate. Since the wavenumber $k$ is zero, these resonances are associated with infinite wavelength $\lambda$. They are not localised and behave like plane waves propagating through the thickness and making the entire surface vibrate in phase.

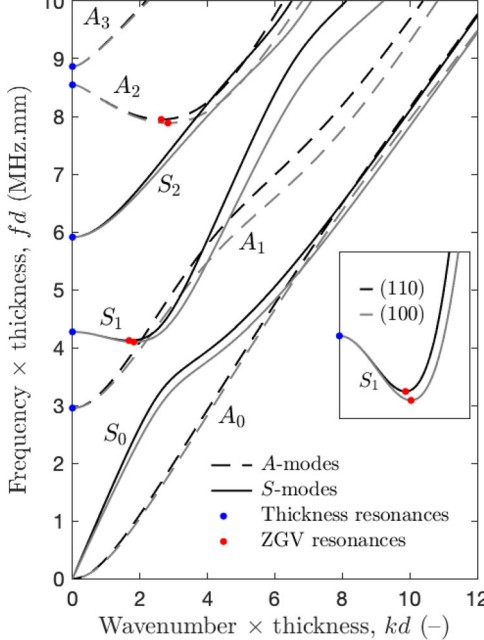

**Fig. 7 Calculated elastic guided modes for bulk silicon.** Dispersion curves in the normalised ($kd$, $fd$)-plane for elastic guided waves propagating in a bulk silicon plate of thickness $d$ along the [100]-axis (grey lines) and the [110]-axis (black lines). Red and blue dots correspond to zero group velocity (ZGV) and thickness resonance frequencies, respectively (the insert displays the frequency-dependent behaviour of ZGV in anisotropic plates).

In addition, some higher-order guided modes (e.g., the first symmetric ($S_1$) and second antisymmetric ($A_2$) modes) exhibit a particular behaviour at frequencies where the group velocity $V_g$ vanishes, while the phase velocity $V_\phi = \omega/k$ remains finite (red dots in Fig. 7). At these ZGV frequencies the energy, which cannot propagate in the plate, is trapped under the source, thus leading to sharp and local resonances. As can be observed in the enlargement of Fig. 7, in contrast to the thickness resonances, the frequency of these particular points depends on the propagation direction[38].

Dispersion curves, along with ZGV and thickness resonances, represent a unique signature of the thickness and mechanical properties of a material, which can be efficiently excited by a laser pulse and optically detected by a sensitive interferometer. In particular, for an isotropic plate of known thickness $d$, the measurement of the two first ZGV resonances provides an accurate estimation of the longitudinal ($V_L$) and shear ($V_T$) bulk wave velocities[36]. Although this appealing feature cannot be straightforwardly extrapolated to anisotropic plates, it has nonetheless been used to provide approximated values for the transversely isotropic properties of Zirconium alloy[77]. This approach will be exploited in this study to deliver a rough estimate of bulk wave velocities in the plate thickness direction for pSi.

**Elastic guided waves measurements**. The LUS setup is an enhanced version of that initially developed by RECENDT GmbH (Linz, Austria). It can be divided into two parts with the sample in between: The left part (in magenta) is responsible for the excitation of EGWs in the sample and the right part (in green) for detecting out-of-plane displacement in the $x_3$-direction (Fig. 8). For pSi, the sample face that was oriented towards the interferometer corresponded to the upper part of the membrane (that with larger pores according to Fig. 2).

EGWs were excited by a passively Q-switched frequency tripled Nd:YAG laser (optical wavelength $\lambda_E = 355$ nm). The pulse repetition was set to 500 Hz. Pulse duration was equal to 350 ps with 25 μJ of energy per pulse and a peak power of 60 kW. The laser was focused by a movable spherical shaped lens with a focal length of 5 cm. The intensity of the laser can be weakened by several neutral density filters. The waves were detected by a stabilised Michelson interferometer equipped with a frequency-doubled Nd:YAG laser (optical wavelength $\lambda_I = 532$ nm). The laser's power was set to 60 mW.) The detection point can be moved along the $x_1$-direction by a motorised linear stage. Typically, measurements were taken at a total travel range of $x_1 = 15$ mm with an incremental step of 10 μm. A piezoelectric driven retroreflector maximised the signal and filtered parasitic long-wave vibrations. The reflector was controlled by a custom piezo controller. The interference signal of the interferometer was detected by an optical diode. The signal is proportional to

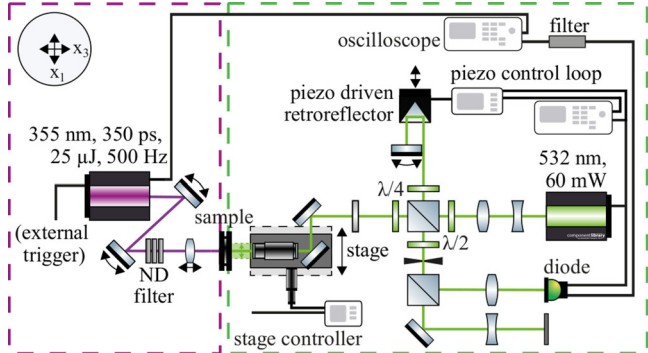

**Fig. 8 Schematic view of the laser ultrasonics setup.** The left side (in magenta) represents the part responsible for the laser excitation, whereas the right side (in green) displays the part used for interferometric waves detection. The Cartesian coordinate system is depicted in the upper left corner.

the phase difference of the two paths. The phase shift caused by the probe is in turn proportional to the probes displacement. The monitor output of the diode was connected to the piezo controller completing the control loop. The high-frequency part of the signal was fed into an oscilloscope with a bandpass filter in-between filtering frequencies lower than 500 kHz. The filter was necessary as the absorbed energy from the laser heated the surrounding air leading to the variation of the index $\Delta n$ in the optical path of the probe beam. This induced an additional phase shift $\Delta\phi_n$ and a large low-frequency voltage that would otherwise saturate the electronics detection unit[78]. A video of a typical signal acquisition is provided in Supplementary Movie 1.

## Data availability

All data needed to evaluate the conclusions in the paper are present in the paper and Supplementary Information. The source data used in this publication is accessible online in the Research Data TUHH collection with the https://doi.org/10.15480/336.3174. Requests for material should be addressed to the corresponding author.

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

## Acknowledgements

The authors are in debt to Andriy V. Kityk (Czestochowa University of Technology) for putting forward the idea of using laser ultrasonics to characterise nanoporous structures and to Dr. Giuseppe Rosi (Université Paris-Est Créteil) for his valuable advises with respect to the numerical modelling. This work was supported by the Deutsche Forschungsgemeinschaft (DFG) within the Collaborative Research Initiative SFB 986 "Tailor-Made Multi-Scale Materials Systems" (Project number 192346071) and the DFG research grant "Dynamic Electrowetting at Nanoporous Surfaces: Switchable Spreading, Imbibition, and Elastocapillarity", Project number 422879465 (SPP 2171). C.P. acknowledges the support of LABEX WIFI (within the French Program "Investments for the Future") under references ANR-10-LABX-24 and ANR-10-IDEX-0001-02 PSL*. N.B. acknowledges the Univ Paris-Est Creteil for the "Support for research for newly appointed Associate Professors".

## Author contributions

M.T., M.B. and P.H. conceived the experiments. M.T. and M.B. performed the material synthesis and the laser ultrasonics experiments. M.T., N.B. and C.P. performed the analysis of the experiments. C.P. addressed the interpretation of the ZGV resonances. N.B. implemented the numerical waveguide model and performed the inverse identification. All authors wrote and proofread the manuscript.

## Funding

## Competing interests

The authors declare no competing interests.
