## [Peer Review File · Nature Communications]

REVIEWER COMMENTS

Reviewer #1 (Remarks to the Author):

The work by Thelen et al. describes a well-detailed study of mechanical properties of mesoporous silicon using laser excited elastic guided waves. Mesoporous silicon is indeed a very interesting material which is exploited in a very diverse applications and has perspectives for many other potential applications. The application areas range from photonics to drug deliver as most diverging examples. Some applications, irrespective of useful properties created by nanostructuring of silicon, are prohibited by the unfavourable mechanical properties. In this regard, their better quantification and understanding is of importance for broad scientific community. In the present work, the authors suggest and successfully demonstrate the potentials of using elastic guided waves as the most robust technique for probing mechanical properties of thin mesoporous silicon chips. I believe that the methodology outlined here and the information obtained will notably contribute to better understanding of nonporous silicon. Because of very diverse fields where mesoporous silicon is used and may potentially be used, the results presented will be of interest for a broad audience, justifying publication in Nature Communications. The technical part also complies with the standards of the journal. The groups contributing have substantial expertise concerning both material and EGW application and analysis. It contains all essential details allowing for the reproduction of the experiments. Altogether I would recommend the work for publication with some minor details to be addressed.

- 1) It turned out that it is essential to take account of the porosity gradient in the samples synthesised in order to . Even though it would not affect the analysis performed, I think it is important to raise a question what the origin of the porosity gradient is. The authors hypothesise "that during the synthesis of pSi the pore walls exposed to HF were isotropically etched, although quantum confinement in the pore walls is likely to inhibit the attack of HF [39]. The upper part of the membrane was exposed longer to the acid and thus cone-shaped pores developed (Figure 3)." On the other hand, it is known that deterioration of the etchant properties with progressive etching leads to the same effect, but with the opposite sign. In this way conical pores, but with narrower parts in contact to the upper etchant bath are formed (see, e.g. Canham, L. Mesoporous Silicon (Springer International Publishing, 2017). Thus, if the latter mechanism is valid, the pore iconicity can be intentionally varied and the results presented independently verified.
- 2) I believe that the surface properties of the mesoporous films used need to be described in greater details. The freshly prepared samples oxidise in air and ultimately the layer formed on the pore walls affects the mechanical properties.

Reviewer #2 (Remarks to the Author):

General Comments

This work describes a new nondestructive method to study the effective elastic properties and filling density of air and fluid filled porous anisotropic materials. The technique relies on the measurement and analysis of the dispersion behavior of laser generated elastic guided waves (EGWs). The measured EGW data exhibited several interesting phenomena including, repulsion at mode crossings, splitting of Rayleigh waves in porous membrane, transition from anisotropy-dominated dispersion in bulk membranes to pore-microstructure-dominated wave dispersion in the nanoporous membrane, etc. The authors provide an exhaustive description of these effects in the context of symmetry breaking in the porous membrane, where symmetry is defined with respect to the mid-plane of the membrane. Furthermore, numerical inverse modeling approaches are explored to predict the effective elastic constants of the porous membrane and the fluid filling density of the samples. The work was a joy to read and provides several new insights on the propagation of elastic waves in nanoporous materials, that may be valuable to fundamental studies

of the coupling of nano-capillary structures and elastic matrices in nanoporous solids. The work also provides some promising new directions for future research connected with tailoring the interaction of guided and Rayleigh waves, based mode repulsion and splitting using nanoporous microstructures. I strongly recommend that the manuscript be accepted for publication after minor revisions.

Specific Comments

1. On Page 6 the authors suggest that due to symmetry reduction, the S_0 and A_0 modes diverge, leading to two Rayleigh waves with different velocities and with propagation paths on different surfaces. Since the measurements were performed on the same surface and along the same path, it is not clear what this statement actually means.

2. In the 2D FFT data presented in Fig. 6, I noticed that the amplitude of the qS_0 mode changed significantly above 2.7 MHz compared to the qA_0 . The same applies to the S_0 and A_0 mode in the bulk silicon data. Since the interferometric probe is mostly sensitive to the out-plane displacement, and the S_0 mode has a dominant in-plane displacement, one would expect a lower displacement sensitivity to the S_0 mode. The 2D FFT data supports this idea below 2.7 MHz, where the amplitude of the A_0 or qA_0 mode is dominant. Was there a sudden change in the polarization of the S_0 or qS_0 mode above 2.7 MHz, and what may be responsible for this change ?

3. On page 7, the mass density at the mid-plane of the membrane is used for the inverse calculation of the stiffness coefficients of the porous silicon membrane. Why was the average density across the membrane thickness not used ?

Reviewer #3 (Remarks to the Author):

The manuscript describes a new approach for the characterization of free standing nanoporous silicon plates using guided elastic waves. The authors use a laser source to generate guided waves in the plate and an optical interferometer is used to detect the displacement of the sample surface. The authors map out the dispersion behavior by measuring the displacement field as a function of distance and taking a two-dimensional Fourier transform. They also detect localized resonances of the membrane associated with zero group velocity modes. An inverse algorithm is proposed and used to estimate the elastic constants of the membrane and a geometric parameter associated with the pore shape. The authors provide experimental results for Si, porous Si, and liquid infused porous Si. While the experimental technique is not novel in itself, the application to nanoporous materials represents an important step. In particular, the authors show that the non-uniformity of the pore distribution results in a repulsion of modes associated with a lack of symmetry of the plate about the mid-plane. The authors also demonstrate that the approach can be used to probe liquid media confined to the pores. The technique may find application in the study of confinement effects and better understanding the mechanics of other porous structures. The paper is well-written and I believe that the work will be of interest to a broad community. I believe that it should be published in Nature Communications after the authors address the concerns listed below.

(a) The authors choose a transversely isotropic model based on the result shown in Fig. 6(c) and (f). I do not believe that this is convincing as the resonance may be, for example, broadened for the porous plate and the two peaks not clearly separated. I suggest that the authors measure the dispersion curve along $[100]$ and compare it directly to the $[110]$ curve or use a line source to compare the resonance in each direction. As the pores appear to be randomly distributed in (001) and the crystal structure is not changed, it seems that a certain degree of anisotropy would be maintained. Perhaps this would be more evident in the higher order modes?

(b) The authors choose to represent the membrane as homogeneous and transversely isotropic, with a density variation perpendicular to the surface. I am not sure that I completely understand the rationale for this. The effective stiffness also varies along $[001]$ and presumably this would lead to some of the observed effects such as mode repulsion. How can this be decoupled from the

density variation? In other words, won't the reported value for beta also be affected by the stiffness distribution?

Minor comments

(c) Typo page 4, just before section B "see section ??"

(d) Page 5 end of first column and beginning of next- consider rephrasing this as a shear wave arrival is not evident to me. Also slope of the curve is inversely proportional to the velocity and steeper for p-Si

(e) Fig. 6 caption- consider not referring to the black peak as an artifact

(f) Page 6, section B1 typo- "sise"

(g) Page 7 in the inverse method- what are the values of P and Q?

Responses to Reviewers

The authors greatly appreciate the Reviewers' efforts to provide comments and constructive criticisms, which have guided us to revise this manuscript. We have carefully followed the Reviewers' suggestions and have made several changes in this revised version. The detailed response to each issue pointed out by the Reviewers (in *italics*) are commented below (in regular typography), with its corresponding page numbers (in **bold**). The corresponding changes are highlighted in **blue** in the revised manuscript.

REVIEWER # 1

General comment:

The work by Thelen et al. describes a well-detailed study of mechanical properties of mesoporous silicon using laser excited elastic guided waves. Mesoporous silicon is indeed a very interesting material which is exploited in a very diverse applications and has perspectives for many other potential applications. The application areas range from photonics to drug deliver as most diverging examples. Some applications, irrespective of useful properties created by nanostructuring of silicon, are prohibited by the unfavourable mechanical properties. In this regard, their better quantification and understanding is of importance for broad scientific community. In the present work, the authors suggest and successfully demonstrate the potentials of using elastic guided waves as the most robust technique for probing mechanical properties of thin mesoporous silicon chips. I believe that the methodology outlined here and the information obtained will notably contribute to better understanding of nonporous silicon. Because of very diverse fields where mesoporous silicon is used and may potentially be used, the results presented will be of interest for a broad audience, justifying publication in Nature Communications. The technical part also complies with the standards of the journal. The groups contributing have substantial expertise concerning both material and EGW application and analysis. It contains all essential details allowing for the reproduction of the experiments. Altogether I would recommend the work for publication with some minor details to be addressed.

Response: We greatly appreciate the positive feedback.

The reviewer does feel that some minor details need to be addressed.

- 1. It turned out that it is essential to take account of the porosity gradient in the samples synthesised in order to . Even though it would not affect the analysis performed, I think it is important to raise a question what the origin of the porosity gradient is. The authors hypothesise “that during the synthesis of pSi the pore walls exposed to HF were isotropically etched, although quantum confinement in the pore walls is likely to inhibit the attack of HF [43]. The upper part of the membrane was exposed longer to the acid and thus cone-shaped pores developed (Figure 3).” On the other hand, it is known that deterioration of the etchant properties with progressive etching leads to the same effect, but with the opposite sign. In this way conical pores, but with narrower parts in contact to the upper etchant bath are formed (see, e.g. Canham, L. Mesoporous Silicon (Springer International Publishing, 2017). Thus, if the latter mechanism is valid, the pore conicity*

can be intentionally varied and the results presented independently verified.

Response: Thank you for pointing this out. Of course, understanding the mechanism behind the porosity gradient is of great importance but would exceed the objectives of this paper. Future works could explore the gradient and its origin further. Nevertheless, our observation has been noticed and interpreted independently from these recent studies [1–4]. For instance, Strehlke et al. proposed that during the synthesis process the sample is not only etched electrochemically but chemically as well, due to the high reactivity of pSi. For the interested reader two additional references have been added to the manuscript.

[1] Strehlke et al., Characterization of thin porous silicon films formed on n^+/p silicon junctions by spectroscopic ellipsometry. *Journal of The Electrochemical Society*, **147**: 636, 2000.

[2] Aroutiounian et al., Calculations of the reflectance of porous silicon and other antireflection coating to silicon solar cells. *Thin Solid Films*, **403–404**: 517–521, 2002.

[3] Ghulinyan et al., Porous silicon free-standing coupled microcavities. *Applied Physics Letters*, **82**, 1550–1552, 2003.

[4] Busch, Optische Untersuchungen zum Kapillarsteigen von Flüssigkeiten in mesoporen Festkörpern. *Master's thesis*, 76–83, Universität des Saarlandes, 2012.

Page 2, column 2, paragraph 1, the manuscript now reads: “Cone-shaped pores and their significant influence on key properties like optical response and capillary pressure have already been evidenced for pSi [39–42].”

2. *I believe that the surface properties of the mesoporous films used need to be described in greater details. The freshly prepared samples oxidise in air and ultimately the layer formed on the pore walls affects the mechanical properties.*

Response: Thank you again for this suggestion. The surface properties of the membrane have now been described in greater detail. Concerning the possible oxidation of the sample's surface, which in turn could lead to the presence of a thin layer on the pore walls, we do not believe that this may alter the effective mechanical properties of the pSi sample significantly, as the thickness of that layer (expected to be only a few nanometres thick) is much smaller than the wavelengths of the elastic guided waves (*i.e.*, these typically range from around 100 μm to 10 mm). Overall, the assessment of oxidation or other factors that may alter the surface properties of the sample remains beyond the scope of this study. Nevertheless, the effects of ageing on the mechanics could be explored in future studies.

Page 3, column 1, paragraph 1, the manuscript now reads: “After the synthesis procedure the sample was rinsed with deionised water and dried in the air for one day before the measurements took place.”

REVIEWER #2

General comments:

This work describes a new nondestructive method to study the effective elastic properties and filling density of air and fluid filled porous anisotropic materials. The technique relies on the measurement and analysis of the dispersion behavior of laser generated elastic guided waves (EGWs). The measured EGW data exhibited several interesting phenomena including, repulsion at mode crossings, splitting of Rayleigh waves in porous membrane, transition from anisotropy-dominated dispersion in bulk membranes to pore-microstructure-dominated wave dispersion in the nanoporous membrane, etc. The authors provide an exhaustive description of these effects in the context of symmetry breaking in the porous membrane, where symmetry is defined with respect to the mid-plane of the membrane. Furthermore, numerical inverse modeling approaches are explored to predict the effective elastic constants of the porous membrane and the fluid filling density of the samples. The work was a joy to read and provides several new insights on the propagation of elastic waves in nanoporous materials, that may be valuable to fundamental studies of the coupling of nano-capillary structures and elastic matrices in nanoporous solids. The work also provides some promising new directions for future research connected with tailoring the interaction of guided and Rayleigh waves, based mode repulsion and splitting using nanoporous microstructures. I strongly recommend that the manuscript be accepted for publication after minor revisions.

Response: We greatly appreciate the positive feedback.

The reviewer asks for some minor revisions.

1. *On Page 6 the authors suggest that due to symmetry reduction, the S_0 and A_0 modes diverge, leading to two Rayleigh waves with different velocities and with propagation paths on different surfaces. Since the measurements were performed on the same surface and along the same path, it is not clear what this statement actually means.*

Response: Thank you for your comment. Rayleigh waves are surface waves, which can typically be observed for large frequency-thickness products, *i.e.*, the region of the dispersion curves where the S_0 and A_0 modes reach their asymptotic behaviour (for a symmetric plate). For an asymmetric plate, the coupling of surface waves disappears when the Rayleigh waves have different phase velocities on each surface. Then the qS_0 mode corresponds to the fastest wave and qA_0 to the slowest one. It is true that the measurements were performed on the same surface (*i.e.*, the one with larger pore opening), and this is exactly the reason why at high frequencies, we only observed the qA_0 mode that corresponds to the slowest Rayleigh wave (recall Fig. 6e). Nevertheless, in order to further support this statement, in the Supplementary Materials A we provided as well an example of measurements on another sample with similar characteristics, for which the measurements were performed on both surfaces (see Fig. 1 below).

To clarify this statement, a sentence has been modified in the manuscript.

Page 5, column 1, paragraph 3 the manuscript now reads: “Owing to this symmetry reduction these two modes now diverge, each one with a different Rayleigh wave velocity

Figure 1: Dispersion spectra derived from LUS measurements performed on both sides of a pSi membrane with similar characteristics to that presented in the manuscript. Zero-order symmetric (qS_0) and antisymmetric (qA_0) modes do not converge to the same Rayleigh wave velocity due to the break in symmetry at around $kd \approx 8$ (red circles) caused by the cone-shaped pores. This results in two different Rayleigh (surface) waves, measured with different intensities on each side (white circles) of the sample. Larger pore opening and hence higher porosity on one side of the sample (a) leads to a lower Rayleigh velocity compared to that on the side with smaller pore opening (b). The combination of both dispersion spectra (c) shows that, in contrast to the lowest order modes, the higher order modes are not impacted by the cone-shaped pores.

associated with a propagation path on a different surface. Concretely, the qS_0 mode converges toward the faster Rayleigh wave on the dense surface and the qA_0 mode toward the slowest Rayleigh wave on the light surface. In Figure 6e, only the qA_0 mode is observed because our measurements were performed on the sample face with larger pore openings. This effect is discussed further in the Supplementary Materials A.”

2. In the 2D FFT data presented in Fig. 6, I noticed that the amplitude of the qS_0 mode changed significantly above 2.7 MHz compared to the qA_0 . The same applies to the S_0 and A_0 mode in the bulk silicon data. Since the interferometric probe is mostly sensitive to the out-plane displacement, and the S_0 mode has a dominant in-plane displacement, one would expect a lower displacement sensitivity to the S_0 mode. The 2D FFT data supports this idea below 2.7 MHz, where the amplitude of the A_0 or qA_0 mode is dominant. Was there a sudden change in the polarization of the S_0 or qS_0 mode above 2.7 MHz, and what may be responsible for this change ?

Response: Thank you for this remark. While it is true that at low frequency the nondispersive S_0 (or qS_0 for pSi) mode is a compression mode with a dominant in-plane displacement, above the “knee” located at 3.3 MHz.mm (or 2.4 MHz.mm for pSi), the out-of-plane displacement of this mode becomes significant. To support this, the displacements at the normalised frequencies $fd = 1.5$ MHz.mm and $fd = 3.7$ MHz.mm are depicted in Fig. 2 for the bulk silicon sample.

3. On page 7, the mass density at the mid-plane of the membrane is used for the inverse calculation of the stiffness coefficients of the porous silicon membrane. Why was the av-

Figure 2: (a) Modelled dispersion curves at low frequencies for the bulk silicon sample; (b) Displacements of the S_0 mode for the plate mode at $fd = 1.5$ MHz.mm (circle in (a)); and (c) Displacements of the S_0 mode in the dispersive region at $fd = 3.7$ MHz.mm (diamond in (a)). In-plane and out-of-plane displacements are displayed in blue and red, respectively. As can be observed, the out-of-plane displacement becomes significant in the dispersive region, thus explaining the higher amplitude for S_0 (or qS_0) in the measured dispersive spectra of Fig. 6b–e.

erage density across the membrane thickness not used?

Response: We think that there is a slight misunderstanding here. Since the density profile $\rho(x_3)$ varies linearly across the thickness, the average mass density across the thickness is equal to the mass density at the mid-plane, *i.e.*, $\rho(d/2) = \rho_0$ according to Eq. (2). Anyway, please note that we now parameterised our waveguide model directly in terms of a varying stiffness-to-weight tensor across the membrane’s thickness instead of a linearly varying density profile (see answer to question 2 raised by Reviewer #3), so that the statement on the average density becomes obsolete. It should be noted, however, that the reported properties and conclusions derived from this model reformulation are not significantly affected by this modification.

REVIEWER #3

General comments:

The manuscript describes a new approach for the characterization of free standing nanoporous silicon plates using guided elastic waves. The authors use a laser source to generate guided waves in the plate and an optical interferometer is used to detect the displacement of the sample surface. The authors map out the dispersion behavior by measuring the displacement field as a function of distance and taking a two-dimensional Fourier transform. They also detect localized resonances of the membrane associated with zero group velocity modes. An inverse algorithm is proposed and used to estimate the elastic constants of the membrane and a geometric parameter associated with the pore shape. The authors provide experimental results for Si, porous Si, and liquid infused porous Si. While the experimental technique is not novel in itself, the application to nanoporous materials represents an important step. In particular, the authors show that the non-uniformity of the pore distribution results in a repulsion of modes associated with a lack of symmetry of the plate about the mid-plane. The authors also demonstrate that the approach can be used to probe liquid media confined to the pores. The technique may find application in the study of confinement effects and better understanding the mechanics of other porous structures. The paper is well-written and I believe that the work will be of interest to a broad community. I believe that it should be published in Nature Communications after the authors address the concerns listed below.

Response: We greatly appreciate the positive feedback and acknowledge the reviewer for pointing out several interesting issues.

The reviewer asks for comments and changes.

- 1. The authors choose a transversely isotropic model based on the result shown in Fig. 6(c) and (f). I do not believe that this is convincing as the resonance may be, for example, broadened for the porous plate and the two peaks not clearly separated. I suggest that the authors measure the dispersion curve along [100] and compare it directly to the [110] curve or use a line source to compare the resonance in each direction. As the pores appear to be randomly distributed in (001) and the crystal structure is not changed, it seems that a certain degree of anisotropy would be maintained. Perhaps this would be more evident in the higher order modes?*

Response: Thank you for this important remark. We agree that a certain degree of anisotropy might persist, despite the presence of the porous network that is preferentially oriented across the membrane's thickness. To address this issue, we now also added inserts depicting the enlargement of the second ZGV peak (*i.e.*, $f_{qA_2A_3}$) for pSi in Fig. 3e–f. As can be observed, both the presence of an additional pseudo cut-off frequency (enlargement in Fig. 3e) and the splitting of the ZGV resonance (enlargement in Fig. 3f) are not as obvious as for the bulk silicon sample. According to [Royer et al., *I2M-Méthodes innovantes en CND*, 2010], the degree of anisotropy can be quantified by evaluating the ratio $\Delta f/f$, which approximately leads to 0.8% and at most 0.3% for the bulk silicon and pSi samples, respectively. In other words, this means that the degree of in-plane anisotropy is reduced for the pSi sample, although the lack of resolution in Fig. 3e–f makes a precise quantification of this reduction difficult.

Figure 3: *Laser-ultrasonic measurements for bulk silicon (upper panels) and nanoporous silicon (lower panels). (a)–(d) Spatio-temporal broadband displacements measured on the samples with the laser-ultrasonic setup; (b)–(e) Dispersion curves in the normalised wavenumber-frequency plane, obtained by 2D Fourier transform of the spatio-temporal signals (for bulk silicon, theoretical guided modes are displayed in white continuous lines for comparison); and (c)–(f) ZGV resonances (the insert depicts the presence of two ZGV resonances for bulk silicon, where the red one is the actual resonance belonging to (110)-axis and the black one reveals the concurrent excitation of the resonance belonging to [100]-axis due to anisotropy).*

However, our measurements allow us to highlight a strong change of the anisotropy in the x_1x_3 -plane. It is well known that, for the bulk silicon sample, the longitudinal bulk wave velocity is equal to $8.43 \text{ mm} \cdot \mu\text{s}$ in the [100]-direction (and [001]-direction for a cubic symmetry), and to $9.13 \text{ mm} \cdot \mu\text{s}$ in the [110]-direction, which is 8% higher. According to our inverse results for pSi, the longitudinal bulk wave velocity is equal to $5.63 \text{ mm} \cdot \mu\text{s}$ in the [001]-direction, and to $4.58 \text{ mm} \cdot \mu\text{s}$ in the [110]-direction, which is 23% lower thus revealing a strong change of the anisotropy. Overall, this means that, although some in-plane anisotropy may remain, the longitudinal bulk wave velocity in the [100]-direction would not be less than $4.2 \text{ mm} \cdot \mu\text{s}$ (assuming a 8% decrease as lower bound). Consequently, the impact of the porous network (for such a porosity level) is much stronger than the potentially remaining in-plane anisotropy, which *a posteriori* justified the choice of a transversely isotropic waveguide model.

Additionally to our observations, Ref. [19] concluded, from a 2D finite element model, that the random structure of the pSi network dominates the macroscopic in-plane stiffness

of the material and the anisotropy of the silicon matrix was negligible. The presented model was based on transmission electron microscopy images of pSi.

Therefore, although a transversely isotropic model might represent only an approximation, the excellent agreement found between the modelled and measured guided modes in Fig. 8a tells us that the reported values for pSi should at least be correct for the [110]-direction. It is true that performing measurements along different propagation directions as in [31, 36] would allow us verifying such hypothesis and being more confident about the exact symmetry class of pSi. At the moment our setup did not allow us to mount a cylindrical lens to obtain a line source but we agree this is something we should do in the future. Moreover, since the investigated pSi sample has been subsequently filled with liquid, the contamination of the pores by the liquid limited further investigations of the very same sample in its “dry configuration”. To account for this limitation concerning the modelling approximation with respect to the degree of anisotropy, some parts of the results (Section III) have been modified and some of the achievements underlined in the discussion (Section IV) have been shaded.

Page 1, abstract, the manuscript now reads: “These experiments reveal that the self-organised formation of 100 billions of parallel nanopores per square centimetre cross section results in a nearly isotropic elasticity perpendicular to the pore axes and an 80% effective stiffness reduction, altogether leading to significant deviations from the cubic anisotropy observed in bulk silicon.”

Page 5, column 2, paragraph 2, the manuscript now reads: “In contrast to the bulk silicon sample, for pSi a splitting of the ZGV resonances cannot be clearly resolved (see the enlargement of Figure 6f). This observation suggests that, although some degree of anisotropy may remain in the x_1x_2 -plane of the plate, for the considered porosity the presence of the pores prevails over the original crystalline nature of bulk silicon (*i.e.*, cubic anisotropy) [19, 54].”

Page 5, column 2, paragraph 5, the manuscript now reads: “Second, based on experimental observations (Figure 6e–f), we hypothesise that the porous network prevails over the crystalline nature of the bulk silicon membrane, so that the mechanical behaviour of pSi can be approximated by a transversely isotropic waveguide model (x_1x_2 being the isotropy plane).”

Page 8, column 2, paragraph 1, the manuscript now reads: “(3) the porous network was assumed to prevail over the crystalline nature of the bulk silicon membrane, and in such a case using a transversely isotropic model was proven to represent a reasonable approximation to identify waveguide properties at the mesostructural scale.”

Page 9, column 2, paragraph 1, the manuscript now reads: “Furthermore, it should be noted that the LUS measurements were performed along the [110]-direction only, thus limiting the investigation of the in-plane anisotropy of the pSi sample. As such, the assumption of a waveguide model with a transversely isotropic mechanical behaviour still demands direct proof, but the obtained inverse results *a posteriori* justified that this approximation is accurate enough to recover waveguide properties along that direction. Future studies are warranted to address the question of whether or not the in-plane anisotropy vanishes for pSi. For instance, measuring the dispersion curves along different

directions in-plane [31] or tracking the ZGV modes as a function of the line source orientation [36] would allow elucidating whether a certain degree of in-plane anisotropy persists and how much.”

2. *The authors choose to represent the membrane as homogeneous and transversely isotropic, with a density variation perpendicular to the surface. I am not sure that I completely understand the rationale for this. The effective stiffness also varies along [001] and presumably this would lead to some of the observed effects such as mode repulsion. How can this be decoupled from the density variation? In other words, won't the reported value for beta also be affected by the stiffness distribution?*

Response: Thank you for this important remark. As the reviewer may know, like for any other ultrasound technique, our laser-based measurements are sensitive to wave velocities, which can, in turn, be related to stiffness coefficients by prescribing the mass density (or vice versa). Therefore, the stiffness coefficients cannot be decoupled from the mass density. Since it is common to relate a variation of porosity to a variation of density (see for instance some works in the bone community [1, 2]), following the schematic of Fig. 3, it resulted quite natural to select a density variation across the thickness. Unfortunately, this resulted in a density gradient pointing towards the wrong direction in the former version of the manuscript. One way to solve this issue would have been to fix the density and make the stiffness coefficients vary across the membrane's thickness. Nevertheless, in order to avoid any misconceptions and to improve the clarity of the paper, we now describe and parameterise our waveguide model directly in terms of a varying stiffness-to-weight tensor, which shows some similarities to Ref. [3]. As these modifications only impact the direction of the bulk wave velocity profiles (see Figure 4 below, which now has been added to the Supplementary Materials C), the excellent to moderate agreement between the modelled and measured data for the dry and liquid-infused case were not significantly impacted. The same applies to the effective properties derived from the optimised model. All the figures and inferred properties have been updated and are highlighted in the manuscript. To clarify this, the most significant modifications related to this model reformulation are summarised below.

Figure 4: *Bulk wave velocity profiles across the membrane's thickness, which result from the model parameterisation in terms of linearly varying stiffness-to-weight ratios: (a) Transverse longitudinal bulk wave velocity, V_L^\perp , (b) Axial longitudinal bulk wave velocity, V_L^\parallel , and (c) Shear bulk wave velocity, V_T .*

[1] Granke et al., Change in porosity is the major determinant of the variation of cortical bone elasticity at the millimeter scale in aged women. *Bone*, **49**: 1020–1026, 2011.

[2] Bernard et al., Elasticity-density and viscoelasticity-density relationships at the tibia mid-diaphysis assessed from resonant ultrasound spectroscopy measurements. *Biomech. Model Mechanobiol.*, **15**: 97–109, 2015.

[3] Tofeldt and Ryden, Zero-group velocity modes in plates with continuous material variation through the thickness. *J. Acoust. Soc. Am.*, **141**: 3302–3311, 2017.

Page 6, column 1, paragraph 1, the manuscript now reads: “Third, the depth-dependent porosity $\phi(x_3)$ (recall Figure 3) turns into continuously varying stiffness coefficients $c_{ij}(x_3)$ and density $\rho(x_3)$ across the thickness. Since the competing impact of stiffness versus density cannot be easily disentangled, we introduce a function $\alpha(x_3)$, which is driven by a factor β that accounts for the cone-shaped pores:

$$\alpha_{ij}(x_3) = \frac{c_{ij}(x_3)}{\rho(x_3)} = 2\alpha_{ij}^0 \left((1 - 2\beta) \frac{x_3}{d} + \beta \right), \quad (1)$$

where $\alpha(x_3)$ represents a linearly varying anisotropic stiffness-to-weight tensor across the thickness, with α^0 being the stiffness-to-weight tensor in the mid-plane of the membrane, *i.e.*, at the position $x_3 = d/2$. It should be noted that $\alpha_{ii}(x_3)$ corresponds to the square of the bulk wave velocities, meaning that, overall, Eq. (2) accounts for continuous variations of bulk wave velocities through the membrane’s thickness. Note also that setting $\beta = 0.5$ leads to the trivial case of cylindrical pores, and therefore to constant bulk wave velocities across the thickness. The density in the mid-plane of the membrane can be defined as

$$\rho(d/2) = \rho_m (1 - \phi(d/2)) + \rho_p \phi(d/2), \quad (2)$$

where ρ_m and ρ_p are the density of the membrane’s matrix and of the material within the pores (dry or liquid-infused pSi), respectively. In this way, the stiffness coefficients in the mid-plane can easily be derived as $c_{ij}(d/2) = \rho(d/2)\alpha_{ij}^0$.”

Page 8, column 1, paragraph 2, the manuscript now reads: “The optimal model parameters $\hat{\theta}$ estimated using the inverse identification procedure were as follows: $\hat{\alpha}_{11}^0 = 21 \text{ mm}^2 \mu\text{s}^{-2}$, $\hat{\alpha}_{33}^0 = 31.7 \text{ mm}^2 \mu\text{s}^{-2}$, $\hat{\alpha}_{13}^0 = 7.6 \text{ mm}^2 \mu\text{s}^{-2}$, $\hat{\alpha}_{55}^0 = 11.1 \text{ mm}^2 \mu\text{s}^{-2}$, and $\hat{\beta} = 0.6$, which in turn lead to the following stiffness coefficients $\hat{c}_{11}(d/2) = 22 \text{ GPa}$, $\hat{c}_{33}(d/2) = 33.2 \text{ GPa}$, $\hat{c}_{13}(d/2) = 8 \text{ GPa}$, and $\hat{c}_{55}(d/2) = 11.6 \text{ GPa}$ in the mid-plane of the membrane. These values were then used to optimise the liquid-infused sample model, leading to an optimal fluid density within the pores equal to $\hat{\rho}_p = 0.675 \text{ g cm}^{-3}$.”

Page 8, column 2, paragraph 1, the manuscript now reads: “(4) the resulting effective stiffness coefficients $\hat{c}_{ij}(d/2)$ in the mid-plane of the membrane are much lower than those of bulk silicon (reduction of $\sim 80\%$).”

Page 9, column 1, paragraph 3, the manuscript now reads: “As a limitation, since our measurements are sensitive to wave velocities, the influence of the effective stiffness cannot be decoupled from that of the effective mass density. Therefore, the impact of the cone-shaped pores was accounted for by considering a continuously varying stiffness-to-weight tensor across the thickness, where all components were assumed to vary following the same linear profile.”

3. *Typo page 4, just before section B “see section ??”*

Response: Thank you, this has been corrected.

4. *Page 5 end of first column and beginning of next- consider rephrasing this as a shear wave arrival is not evident to me. Also slope of the curve is inversely proportional to the velocity and steeper for p-Si*

Response: Thank you for this suggestion. The description was indeed misleading and has been rephrased.

Page 5, column 1, paragraph 1 the manuscript now reads: “Before the low frequency oscillation wavefronts of higher frequencies can be spotted in the submicroseconds range. As can be observed, the slope of these wavefronts are much steeper for pSi than for bulk silicon, thus indicating lower wave velocities for the porous sample.”

5. *Fig. 6 caption- consider not referring to the black peak as an artefact*

Response: Thank you, this part of the caption has been modified.

Page 6, caption of Fig. 6, the manuscript now reads: “[...] and the black one reveals the concurrent excitation of the resonance belonging to [100]-axis due to anisotropy”

6. *Page 6, section B1 typo- “sise”*

Response: Thank you, this has been corrected.

7. *Page 7 in the inverse method- what are the values of P and Q?*

Response: These values are equal to the size of the measured dispersion spectra, *i.e.*, 3670×4427 for the bulk silicon sample and 4194×8000 for the pSi sample. These differences can be explained as follows: the steps in frequency df and wavenumber dk for calculating the 2-D Fourier transform were kept constant, but the explored range of frequency and wavenumber are different from one sample to another.

Finally, we again thank the referees for the critical reviewing and their overall very positive assessment of our manuscript. We think the constructive criticisms and suggestions helped us to improve and clarify the content. We hope that our revised manuscript is now suitable for publication in Nature Communications.

On behalf of all co-authors,

Yours Sincerely,

REVIEWERS' COMMENTS

Reviewer #1 (Remarks to the Author):

Dear Editor,

I agree with the revisions made and can only suggest the publication.

Reviewer #2 (Remarks to the Author):

The authors have thoroughly addressed all my concerns. I recommend that the manuscript be published without further revisions. It is great work.

Reviewer #3 (Remarks to the Author):

The authors have done an excellent job in addressing all of the concerns in my original review. I believe that the revised manuscript will be of broad interest to the scientific community and I am therefore happy to recommend publication in Nature Communications.